# WFS1 functions in ER export of vesicular cargo proteins in pancreatic β-cells

Linlin Wang[1], Hongyang Liu[2], Xiaofei Zhang[3,4], Eli Song [2,3], You Wang[2], Tao Xu [1,2,3,5✉] & Zonghong Li [1✉]

The sorting of soluble secretory proteins from the endoplasmic reticulum (ER) to the Golgi complex is mediated by coat protein complex II (COPII) vesicles and thought to required specific ER membrane cargo-receptor proteins. However, these receptors remain largely unknown. Herein, we show that ER to Golgi transfer of vesicular cargo proteins requires WFS1, an ER-associated membrane protein whose loss of function leads to Wolfram syndrome. Mechanistically, WFS1 directly binds to vesicular cargo proteins including proinsulin via its ER luminal C-terminal segment, whereas pathogenic mutations within this region disrupt the interaction. The specific ER export signal encoded in the cytosolic N-terminal segment of WFS1 is recognized by the COPII subunit SEC24, generating mature COPII vesicles that traffic to the Golgi complex. WFS1 deficiency leads to abnormal accumulation of proinsulin in the ER, impeding the proinsulin processing as well as insulin secretion. This work identifies a vesicular cargo receptor for ER export and suggests that impaired peptide hormone transport underlies diabetes resulting from pathogenic *WFS1* mutations.

[1] Guangzhou Laboratory, Guangzhou, China. [2] National Laboratory of Biomacromolecules, CAS Center for Excellence in Biomacromolecules, Institute of Biophysics, Chinese Academy of Sciences, Beijing, China. [3] College of Life Sciences, University of Chinese Academy of Sciences, Beijing, China. [4] Key Laboratory of Molecular Biophysics of the Ministry of Education, College of Life Science and Technology, Huazhong University of Science and Technology, Wuhan, China. [5] Shandong First Medical University & Shandong Academy of Medical Sciences, Jinan, China. ✉email: xutao@ibp.ac.cn; li_zonghong@grmh-gdl.cn

The biogenesis and transport of vesicular cargo proteins such as insulin are critical for the maintenance of blood glucose homeostasis, and defects in the trafficking of the vesicular cargo proteins are linked to the onset of diabetes disease[1,2]. Vesicular cargo proteins are synthesized in the endoplasmic reticulum (ER) and transported to the Golgi complex for further processing before being transferred to the dense core vesicles (DCVs) in pancreatic β-cells[3]. The early sorting step from ER-to-Golgi complex is driven by coat protein complex II (COPII) vesicles, which are composed of the SAR1 GTPase, SEC23/SEC24 heterodimers, and SEC13/SEC31 heterotetramers[4]. Secreted cargoes are incorporated into COPII vesicles by two mechanisms, "cargo capture" and "bulk flow"[5]. Cargo capture involves receptor-mediated ER export of proteins, in contrast to bulk flow, by which cargoes enter COPII vesicles through passive diffusion[5]. Cargo receptors are thought to accelerate the ER export of selective proteins by concentrating these cargoes into COPII vesicles. For example, deficiency of the lipoprotein receptor SURF4 depleted plasma lipids to near-zero in mice[6]. However, only limited cargo receptors have been characterized for mammalian secreted proteins thus far[6–8]. Whether the cargo receptor is involved in ER export of vesicular cargo proteins, especially the proinsulin is still unclear.

The WFS1 gene was identified as a major causative locus for Wolfram syndrome, which is a monogenic form of diabetes and neurodegeneration characterized by juvenile diabetes, optic atrophy, and deafness[9,10]. Currently, more than 100 rare variants of the WFS1 gene have been linked to the juvenile-onset diabetes associated with Wolfram syndrome[11]. In addition, two common variants are strongly associated with susceptibility to type 2 diabetes[12]. Previous studies have shown that the Wfs1 gene is highly expressed in pancreatic islets and the brain[13,14]. WFS1 was shown to be involved in the maintenance of ER calcium homeostasis in pancreatic β-cells[15]. In addition, the interruption of WFS1 resulted in an ER stress response, leading to apoptosis of pancreatic β-cells[13,16–18].

Here, we show that WFS1 is a vesicular cargo protein receptor for ER export, whose ER luminal C-terminal segment directly interacts with the vesicular cargo proteins. The cytosolic N-terminal segment is recognized by the COPII subunit SEC24, finally forming the COPII vesicle and trafficking to the Golgi complex. Deficiency of WFS1 led to abnormal accumulation of proinsulin in the ER, thus impeding proinsulin processing and insulin secretion.

## Results

**WFS1 deficiency impairs proinsulin trafficking from the ER to the Golgi in vitro and in vivo.** During the exploration of the function of WFS1 in pancreatic β-cells, we found an interesting phenomenon in which the distribution of proinsulin was largely different in the scrambled (control) INS1 cells compared with Wfs1 knockdown INS1 cells (shWfs1) with stable expression of a short hairpin RNA specifically targeting Wfs1 (Fig. 1a, b). Moreover, the knockdown of Wfs1 caused a significantly increased ratio of proinsulin to insulin compared with that of the scrambled cells (Fig. 1c), indicating that the delivery of proinsulin to the Golgi is severely impaired. Immunostaining analysis showed that the proinsulin was partially colocalized with ER and Golgi markers in the scrambled INS1 cells (Fig. 1d–k), indicating that it was distributed throughout the transport pathway. However, proinsulin was mainly colocalized with ER marker (Fig. 1d–g), and lost most of the colocalization pattern with Golgi marker in the shWfs1 cells (Fig. 1h–k), suggesting that WFS1 deficiency impairs proinsulin trafficking from the ER to the Golgi.

To further explore whether the WFS1 is required for ER export of proinsulin in vivo, we generated whole-body Wfs1-knockout (KO) mice via the CRISPR-Cas9 technique. The Wfs1 KO mice showed efficient depletion of the WFS1 protein in pancreatic islets (Supplementary Fig. 1a), without evidence of CRISPR-induced off-target indels. Compared with the wild-type (WT) littermates, the Wfs1 KO mice had lower body weights and higher fasting blood glucose levels (Supplementary Fig. 1b, c). In addition, the Wfs1 KO mice exhibited an impaired glucose tolerance, diminished islet size and abnormal islet morphology (Supplementary Fig. 1d–g), which are consistent with previous models[16,17]. Furthermore, glucose-stimulated insulin secretion was impaired in the islets isolated from the Wfs1 KO mice (Supplementary Fig. 1h), implying that WFS1 deficiency affects the function of pancreatic β-cells. Consistent with the in vitro results, proinsulin was mainly colocalized with ER markers, and lost most of the colocalization pattern with Golgi markers in the KO mouse β-cells in comparison to the WT mouse β-cells (Fig. 2a–h), further confirming that WFS1 is required for ER export of vesicular cargo proteins. In addition, WFS1 deficiency induced a significantly increased ratio of proinsulin to insulin, although both of the proinsulin and insulin levels were decreased (Fig. 2i–l). Taken together, these results suggest that WFS1 affects the proinsulin trafficking from the ER to the Golgi complex and hence subsequent proinsulin processing.

**WFS1 directly interacts with vesicular cargo proteins.** Since WFS1 is an ER-associated membrane protein[14], we tested the hypothesis that WFS1 could be a receptor mediating proinsulin trafficking from the ER to the Golgi. We first explored whether WFS1 can directly interact with the proinsulin protein. We employed a bimolecular fluorescence complementation (BiFC) system based on a split yellow fluorescent protein (YFP) variant to test the protein interaction in live cells[19]. Pairwise expression of the WFS1 tagged with N-terminal segment of YFP (nYFP) and the proinsulin tagged with C-terminal segment of YFP (cYFP) plasmids in HEK-293T cells showed a strong fluorescent signal, indicating a direct interaction between the WFS1 and proinsulin proteins. In contrast, no fluorescent signal was detected when pairwise expression of the control ER membrane protein nYFP-TMED9 and proinsulin-cYFP (Fig. 3a), confirming the specificity of this assay. It has been reported that the ER membrane cargo receptor could regulate the transport of a group of proteins with similar functions[6,7,20]. Therefore, we investigated whether WFS1 could interact with other vesicular cargo proteins. Surprisingly, the BiFC results showed that WFS1 interacted with four out of eleven vesicular cargo proteins, including proinsulin, NPY, CPE, and SCG5 in HEK-293T and INS1 cells (Fig. 3a, b and Supplementary Figs. 2 and 3a). These interactions between WFS1 and cargo proteins were further confirmed via the proximity ligation assay (PLA) (Fig. 3c). In addition, the co-immunoprecipitation (IP) results showed that WFS1 protein could pull down the proinsulin, NPY, CPE, and SCG5 proteins, but not the negative control protein GCG, in HEK-293T cells (Fig. 3d). Moreover, the interactions of WFS1 with endogenous cargo proteins were further confirmed by IP in INS1 cells (Supplementary Fig. 3b). Taken together, these results indicate that WFS1 could directly interact with a group of vesicular cargo proteins.

**WFS1 traffics from the ER to the Golgi.** Then, we explored whether WFS1 can be transported from the ER to the Golgi. First, we explored the localization of WFS1 in the isolated fractions of INS1 cells by density gradient centrifugation. As observed in Fig. 4a, WFS1 partly colocalized with the ER marker calnexin and the Golgi marker GM130, indicating that WFS1 is distributed in

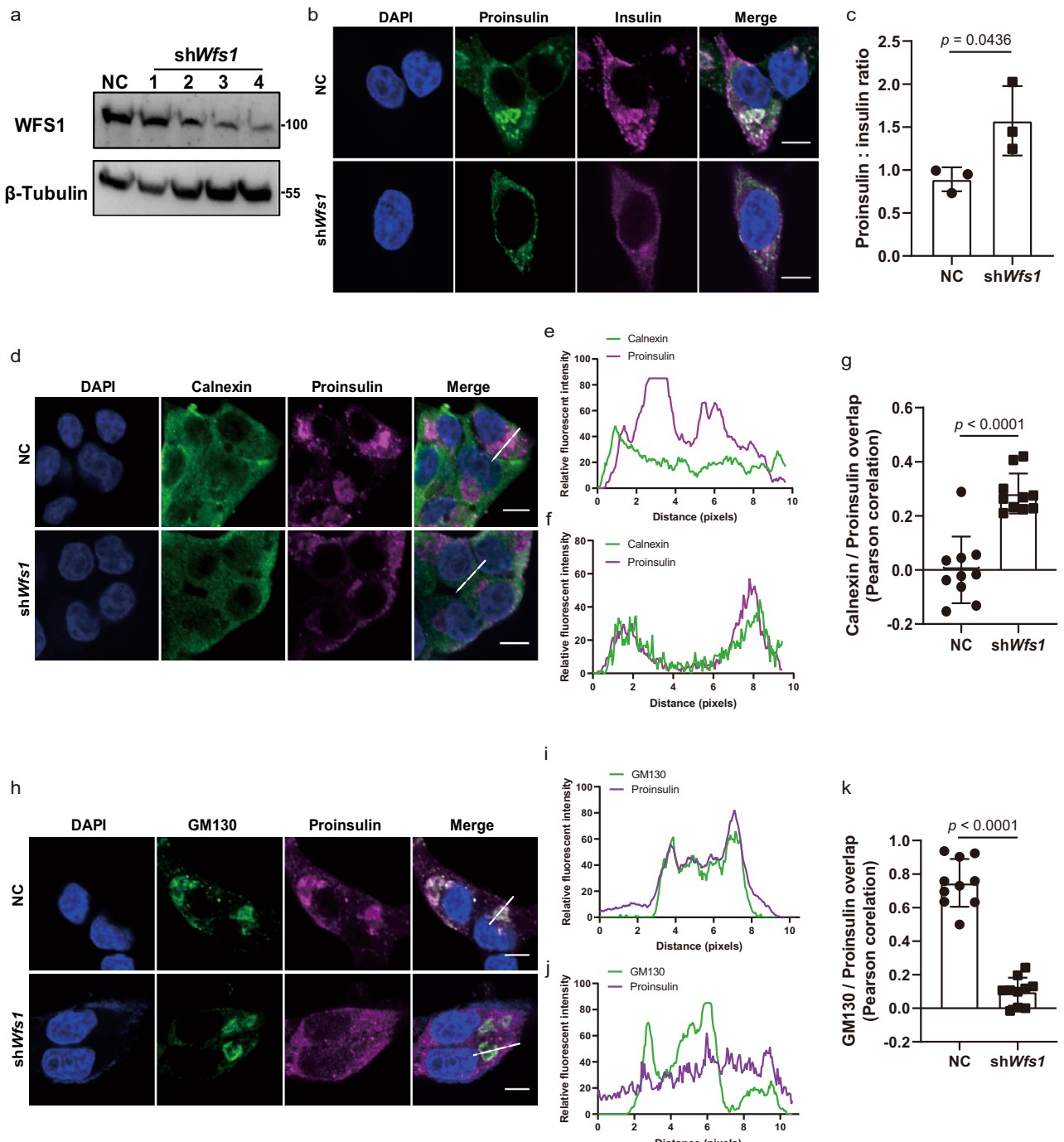

**Fig. 1 Proinsulin translocation from the ER to the Golgi is impaired in sh*Wfs1* cells. a** IB analysis of WFS1 protein in INS1 cells stably expressing the scrambled (NC) and four shRNA plasmids targeting *Wfs1*; β-tubulin was used as the loading control. The fourth clones were used for the following study. *n* = 3 independent experiments. **b** Representative images of immunofluorescence staining of proinsulin and insulin in the WT and sh*Wfs1* INS1 cells. The WT and sh*Wfs1* INS1 cells were immunostained with anti-proinsulin and anti-insulin primary antibodies, followed by Alexa Fluor-conjugated secondary antibodies. Scale bar, 5 μm. **c** The ratio of the fluorescence intensity of proinsulin to insulin was quantified by ImageJ software, *n* = 3 independent experiments. **d**–**k** Confocal microscopy analysis of colocalization of proinsulin with calnexin (ER marker, **d**–**g**) or GM130 (Golgi marker, **h**–**k**) in the WT and sh*Wfs1* INS1 cells. Trace outline is used for line-scan (white dashed line) analysis of the relative fluorescence intensities of proinsulin with calnexin or GM130 signals. Signal overlap is quantified by Pearson correlation analysis. *n* = 3 independent experiments, *n* = 10 independent images quantified. Scale bar, 5 μm. All the data are presented as mean ± s.e.m. *p* < 0.05, significant, using a two-tailed Student's *t*-test. Source data are provided as a Source Data file.

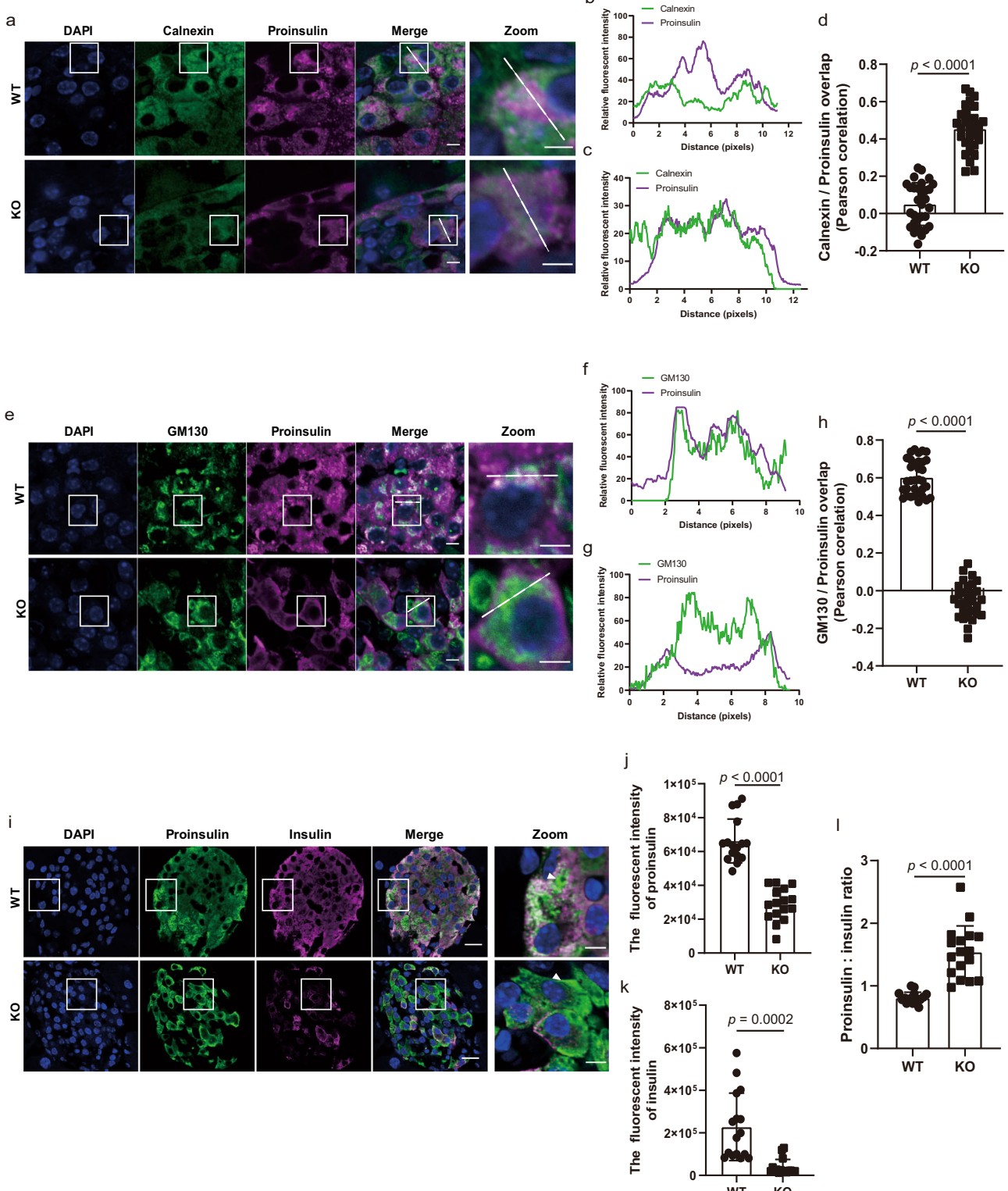

**Fig. 2 Proinsulin translocation from the ER to the Golgi is impaired in *Wfs1* KO mice. a–h** Confocal microscopy analysis of colocalization of proinsulin with calnexin (ER marker, **a–d**) or GM130 (Golgi marker, **e–h**) in pancreatic sections of the WT and *Wfs1* KO mice. Trace outline is used for line-scan (white dashed line) analysis of the relative fluorescence intensity of proinsulin with calnexin or GM130 signals. Signal overlap was quantified by Pearson correlation analysis. $n = 3$ independent experiments, $n = 30$ independent images quantified. Scale bar, 5 µm. Scale bar for zoom figures, 3 µm. **i** Confocal microscopy of proinsulin and insulin in pancreatic sections of the WT and *Wfs1* KO mice. Pancreatic sections were immunostained with anti-proinsulin and anti-insulin primary antibodies, followed by Alexa Fluor-conjugated secondary antibodies. Scale bar, 10 µm. Scale bar for zoom figures, 5 µm. **j–k**, The fluorescence intensities of proinsulin (**j**) and insulin (**k**) were quantified by the ImageJ software. **l** The ratio of fluorescence intensity of proinsulin to insulin. $n = 3$ independent experiments, $n = 16$ independent images quantified. All the data are presented as mean ± s.e.m. $p < 0.05$, significant, using a two-tailed Student's *t*-test. Source data are provided as a Source Data file.

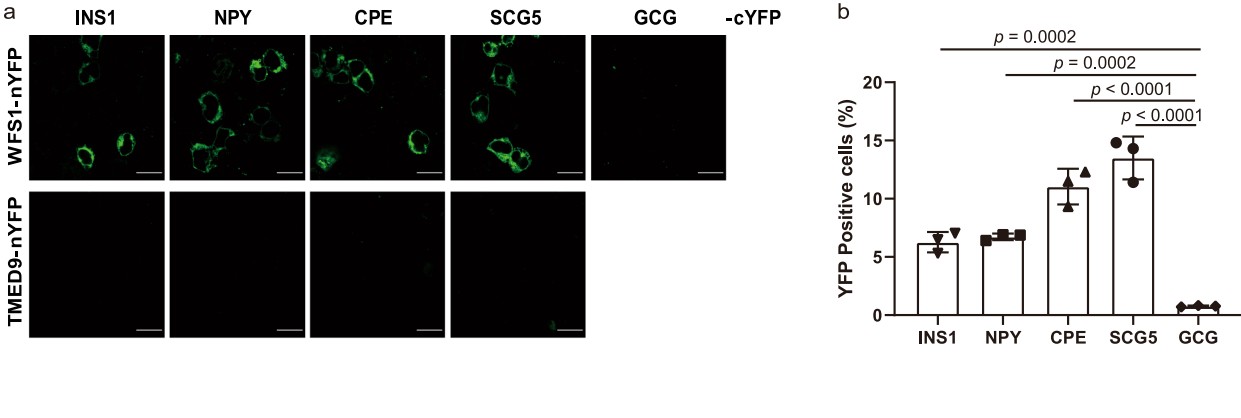

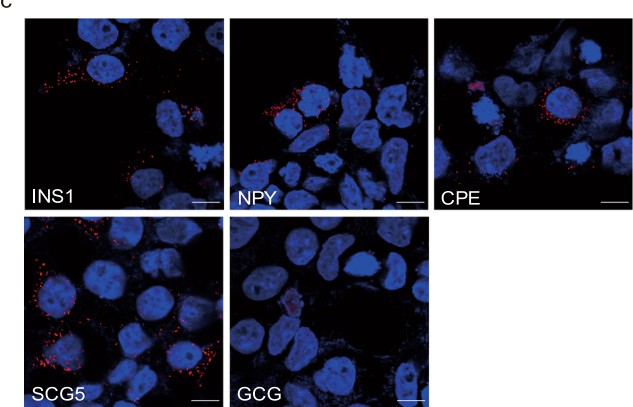

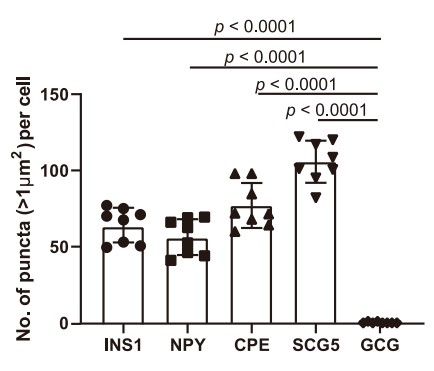

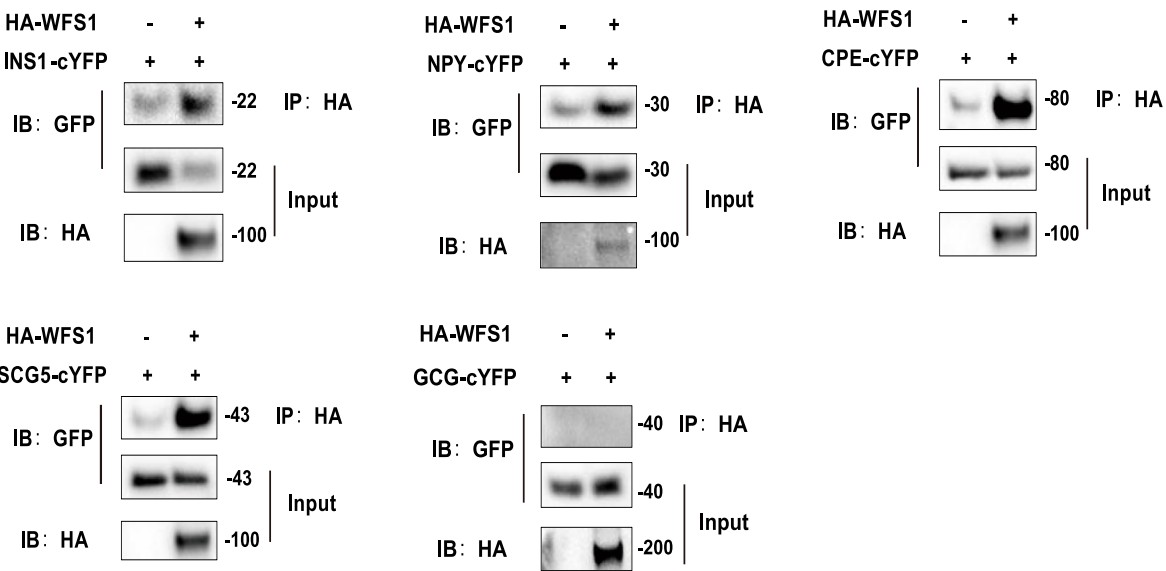

the ER and Golgi. Then, the translocation of WFS1 from the ER to the Golgi was analyzed using a fluorescence microscope. The results showed that WFS1 mainly colocalized with ER markers in HEK-293T and INS1 cells under normal conditions (Fig. 4b and Supplementary Fig. 4a), consistent with WFS1 as an ER membrane protein. Low-temperature (30 °C) treatment, which blocks the secreted protein budding from the Golgi[21,22], increased the

localization of WFS1 to the Golgi complex (Fig. 4b–i and Supplementary Fig. 4). Furthermore, the BiFC results showed that the WFS1 could form a homodimer and that the WFS1 homodimer mostly colocalized with the Golgi complex after low-temperature treatment (Supplementary Fig. 5a), and then trafficked to the plasma membrane after temperature recovery to 37 °C (Supplementary Fig. 5b). These results strongly suggest that WFS1 is a

**Fig. 3 WFS1 interacts with vesicular cargo proteins. a** Representative live imaging of reconstituted BiFC fluorescence between WFS1-tagged nYFP or nYFP-tagged TMED9 and vesicular cargo protein (INS1, NPY, CPE, SCG5, and negative control GCG)-tagged cYFP. Green fluorescence shows reconstitution of YFP as an indicator of protein–protein interactions. Scale bar, 10 μm. **b** The interaction between WFS1 and vesicular cargo proteins evaluated by BiFC, and then quantified by flow cytometry. $n = 3$ independent experiments. **c** Representative imaging of the interaction between WFS1 and vesicular cargo proteins by PLA. PLA is a powerful tool that allows in situ detection of protein interactions with single molecule resolution. HEK-293T cells expressing HA-tagged WFS1 and vesicular cargo protein-tagged cYFP were fixed, and then incubated with mouse anti-HA and rabbit anti-GFP antibody for 1 h at 37 °C, PLA was then performed using anti-rabbit PLUS and anti-mouse MINUS PLA probes. The red puncta represent the interaction between one WFS1 protein and one cargo protein. The puncta number per cell was quantified by ImageJ software. $n = 3$ independent experiments, $n = 8$ independent images quantified. Scale bar, 10 μm. **d** Co-IP analysis of WFS1 and vesicular cargo proteins. Proteins were transiently expressed in HEK-293T cells, and immunoprecipitates pulled down by HA antibody were analyzed by IB with the indicated antibodies. GCG was used as a negative control. Input represents 5% of the total cell extract used for immunoprecipitation. Molecular weights are in kDa. $n = 3$ independent experiments. All the data are presented as mean ± s.e.m. $p < 0.05$, significant, using a two-tailed Student's $t$-test. Source data are provided as a Source Data file.

membrane protein that traffics from the ER to the plasma membrane. In addition, previous studies showed that WFS1 localizes to secretory granules in neuronal cells and pancreatic β-cells[23,24], further confirming that WFS1 is a trafficking protein.

**WFS1 is recognized by the COPII vesicle subunit SEC24 via its cytosolic N-terminal segment.** We next investigated the association of WFS1 with the COPII complex, the coatomer of vesicles involved in ER-to-Golgi anterograde trafficking[25]. COPII interacts with the cargo receptors via its SEC24 subunits, which recognize the specific cytosolic ER export signals within the protein sequence of the cargo receptor[26]. The cytosolic N-terminal tail of WFS1 contains two motifs, [158]ENE and [169]ETD, which overlap with the diacidic ER export signal[27] (Fig. 5a). Interestingly, the two motifs are disrupted by two known pathogenic mutations (E158K and E169K) that are linked to Wolfram syndrome[11]. Lysates derived from HEK-293T cells were subjected to WFS1 pull down using purified recombinant Sumo-tagged SEC24A, SEC24B, SEC24C, SEC23A, and SEC23B proteins. The immunoblotting (IB) results revealed that WFS1 interacts with SEC24A, SEC24B, and SEC24C but not with SEC23A and SEC23B (Fig. 5b). These interactions were further confirmed by PLA analysis (Fig. 5c). Conversely, the N-terminal segment of WFS1 pulled down the SEC24A, SEC24B, and SEC24C subunits, suggesting that the interaction is mediated by the N-terminal segment of WFS1 (Fig. 5d). Furthermore, mutagenesis of either of the two diacidic ER export signals (E158K and E169K) within the N-terminal segment abolished the interaction between WFS1 and SEC24A (Fig. 5e, f). Moreover, the truncated N-terminal segment (ΔNT) and two pathogenic mutants (E158K and E169K) of WFS1 totally disrupt its ER localization (Fig. 6), suggesting that the cytosolic N-terminal segment is required for the correct localization of WFS1. Taken together, these results indicate that the cytosolic N-terminal segment of WFS1 is required for the interaction with the COPII subunit SEC24, and certain pathogenic mutations within this region disrupt this interaction.

**WFS1 recognizes the vesicular cargo proteins via its luminal C-terminal segment.** WFS1 is a multispan membrane protein consisting of a long luminal C-terminal segment and four short luminal loops[28]. Additionally, most pathogenic mutations mapped within the luminal C-terminal segment, suggesting that the C-terminal segment plays a vital role in the function of WFS1[11]. We propose that the luminal C-terminal segment could be involved in the interaction of WFS1 with vesicular cargo proteins. To verify this hypothesis, we generated a truncated C-terminal segment (ΔCT) and four pathogenic mutations (G695V, P724L, E809K, and E830A) within the C-terminal segment WFS1 and used them in interaction assays with vesicular cargo proteins. The

confocal microscopy results showed colocalization of the five WFS1 mutants with the ER marker (Fig. 6), suggesting that the C-terminal segment is not essential for its ER localization. In contrast, the BiFC results showed that deletion of the C-terminal segment severely disrupted the interaction of WFS1 with vesicular cargo proteins. Consistently, the four pathogenic mutations also significantly diminished the interactions (Fig. 7a), suggesting that the C-terminal segment of WFS1 is required for its interaction with cargo proteins. In particular, the E830A mutation severely disrupted the interaction, indicating that this site plays a vital role in the recognition of cargo protein. These results were further confirmed by PLA analysis (Fig. 7b). In addition, to illustrate the effect of mutant WFS1 on the distribution of proinsulin, we examined whether overexpression of the WT and E830A mutant *Wfs1* could restore the proinsulin distribution in sh*Wfs1* cells. The results showed that overexpression of WT *Wfs1* could increase the localization of proinsulin to the Golgi complex, which is similar to the scrambled cells. In contrast, overexpression of the E830A mutant *Wfs1* failed to restore the proinsulin distribution (Supplementary Fig. 6). Taken together, the luminal C-terminal segment of WFS1 is required for the recognition of cargo proteins and several pathogenic mutations within the C-terminal segment disrupt this recognition.

## Discussion

Vesicular peptide hormones are synthesized in the ER, traffic through the Golgi complex, and then sorted into the DCVs in pancreatic β-cells[3]. Previous studies were mainly focused on the post-Golgi processes by which cargo proteins are sorted from the trans-Golgi to DCVs[29,30]. Two models have been proposed for DCV cargo sorting in post-Golgi processes. The "sorting by entry" model proposes that DCV cargoes selectively enter nascent DCVs at the trans-Golgi network, whereby the membrane-bound form CPE is thought to act as a sorting receptor for secretory hormones, such as insulin[31,32]. The "sorting by exit" model proposes that sorting occurs by the immature DCV removal of non-DCV cargos and retention of mature DCV cargoes, whereby the adaptor protein AP-1, coiled-coil protein CCDC186 and endosome-associated recycling protein (EARP) complex may be involved in controlling the post-Golgi retention of cargoes in mature DCVs[33,34]. However, the early sorting step from the ER to the Golgi complex remains unclear. Our results reveal an unanticipated trafficking pathway for the vesicular cargo proteins requiring the specific receptor WFS1 to incorporate into the COPII vesicles and then transport them from the ER to the Golgi complex (Fig. 7c). It has been reported that the ER membrane receptors can mediate ER export of different kinds of cargoes, e.g., CLN8 mediates ER export of the lysosomal enzymes[7] and SURF4 mediates ER export of the lipoproteins[6]. Our results identify a cargo receptor for ER export of secretory peptide hormones.

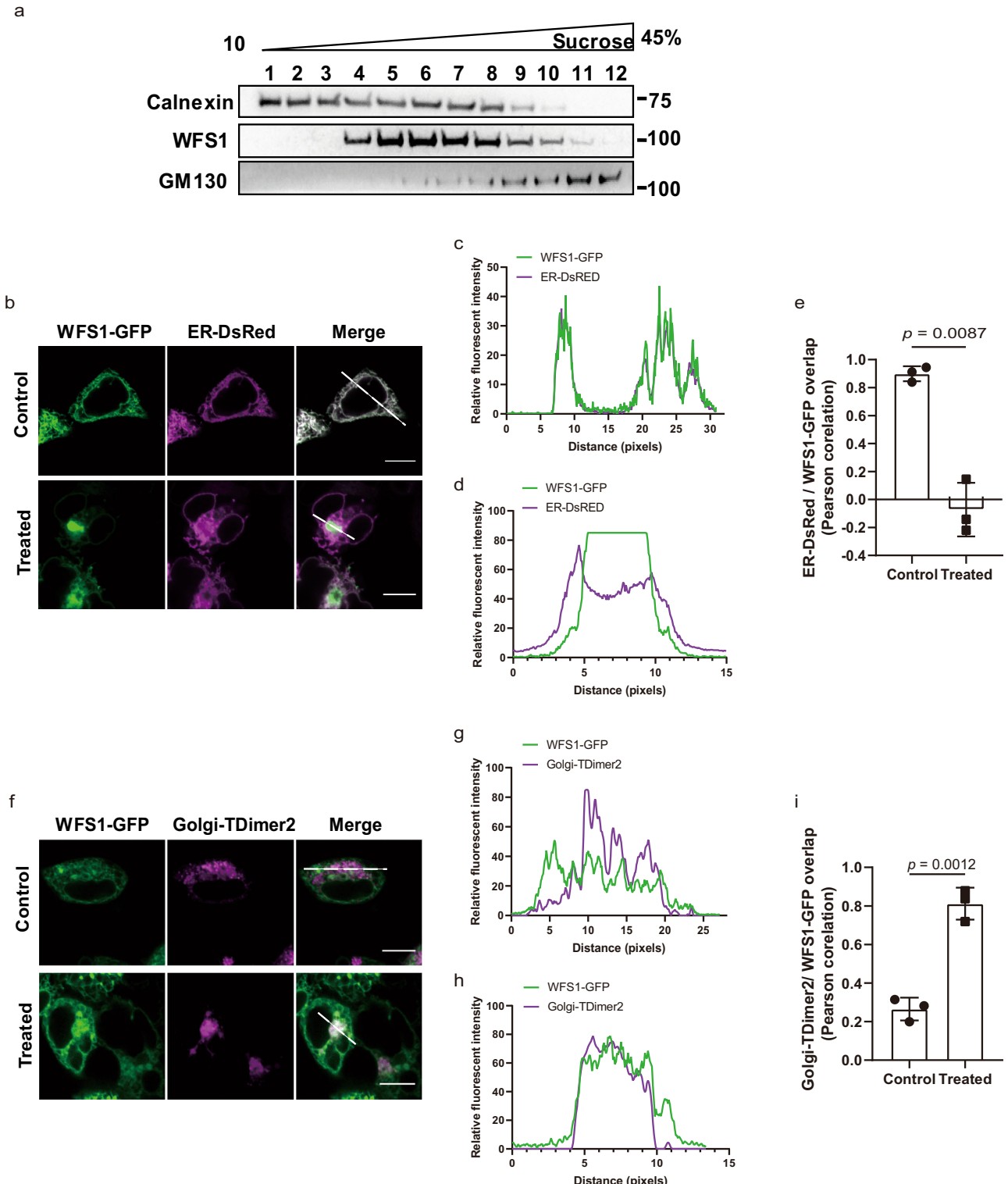

**Fig. 4 WFS1 traffics from the ER to the Golgi. a** WFS1 distributes in the ER and Golgi fractions. Lysates from INS1 cells were subjected to sucrose gradient centrifugation and IB. ER fractions were marked by calnexin and Golgi fractions were marked by GM130. Molecular weights are in kDa. $n = 3$ independent experiments. **b–e** Confocal microscopy analysis of colocalization of WFS1-GFP and ER-Dsred (KDEL, ER marker) in HEK-293T cells treated with normal (37 °C) or low temperature (30 °C) (**b**). Trace outline is used for line-scan (white dashed line) analysis of the relative fluorescence intensity of WFS1 and ER signals treated with normal (37 °C, **c**) or low temperature (30 °C, **d**). Signal overlap was quantified by Pearson correlation analysis of $n = 3$ independent experiments (**e**). Scale bar, 5 µm. **f–i** Confocal microscopy analysis of colocalization of WFS1-GFP and Golgi-TDimer2 (Golgi marker) in HEK-293T cells treated with normal (37 °C) or low temperature (30 °C) (**f**). Trace outline is used for line-scan (white dashed line) analysis of the relative fluorescence intensity of WFS1 and Golgi signals treated with normal (37 °C, **g**) or low temperature (30 °C, **h**). Signal overlap was quantified by Pearson correlation analysis of $n = 3$ independent experiments (**i**). Scale bar, 5 µm. All the data are presented as mean ± s.e.m. $p < 0.05$, significant, using a two-tailed Student's $t$-test. Source data are provided as a Source Data file.

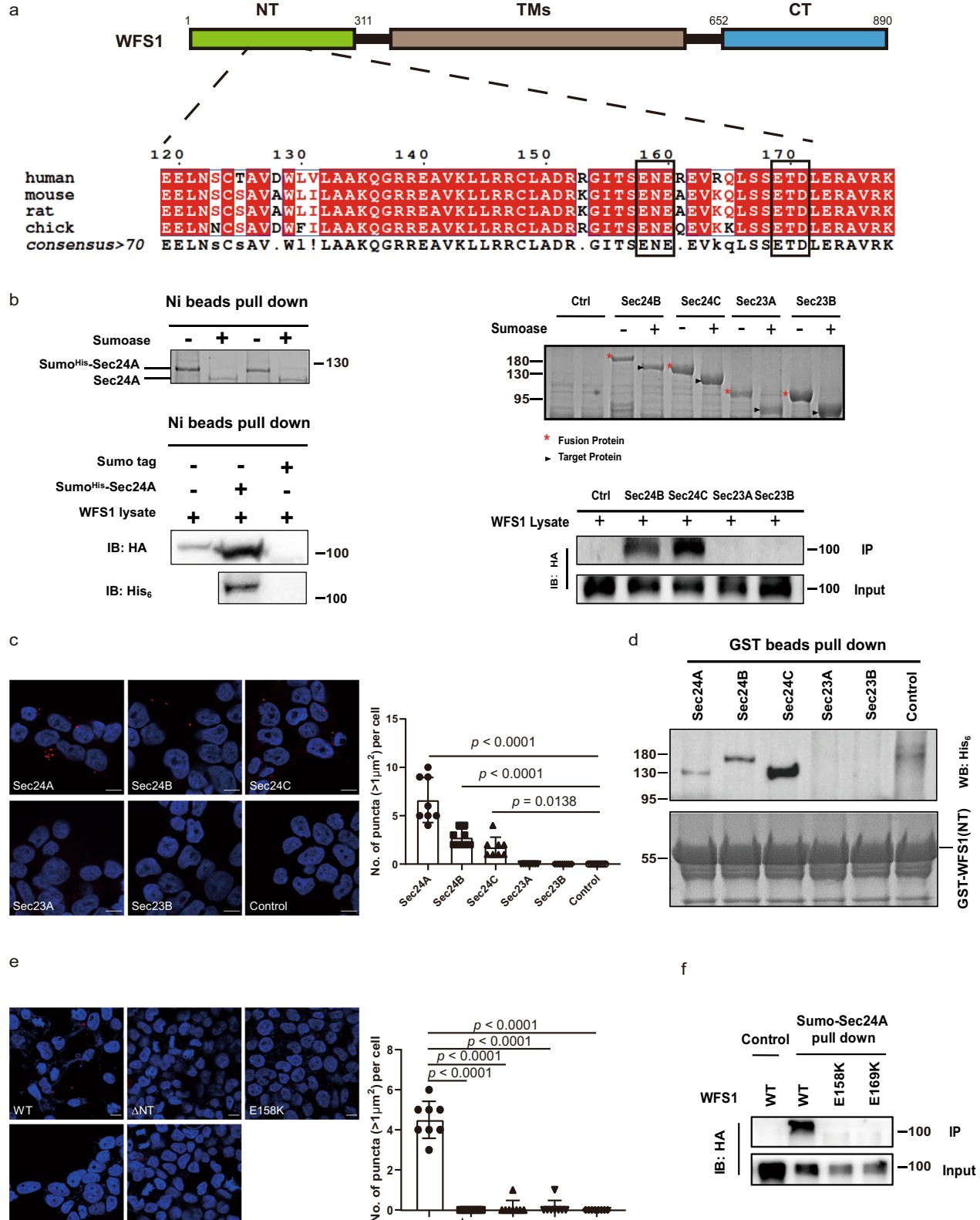

Furthermore, our results propose a potential molecular mechanism underlying the diabetes association with Wolfram syndrome induced by *WFS1* pathogenic mutations. Previous studies have shown that WFS1 deficiency induces an increased ER stress response in pancreatic β-cells, eventually causing the progressive β-cell loss[13,16,17]. However, how the increased ER stress response was induced by WFS1 deficiency remains unclear.

A study reported that WFS1 regulates ER stress through upregulation of ATF6α protein expression and hyperactivation of ATF6α signaling[18]. But the increased ER stress response was only observed in pancreatic β-cells, not in heart, skeletal muscle, or brown adipose tissues from the WFS1-deficient mice[13], indicating a specific function of WFS1 in pancreatic β-cells. Our data showed that the WFS1 is a receptor mediated the secretory

**Fig. 5 WFS1 is recognized by COPII vesicle subunit SEC24 via its cytosolic N-terminal segment. a** Alignment of two putative diacidic ER export motifs (labeled with rectangles) of WFS1 from multiple organisms. Two known pathogenic mutations (E158K and E169K) disrupt these motifs. **b** In vitro pull-down analysis of the interaction between the WFS1 protein and COPII vesicle subunits. Lysates from HEK-293T cells expressing HA-tagged WFS1 proteins were subjected to pull down with purified Sumo$^{His}$-tagged COPII vesicle subunits. IB analysis of pull-down samples showing WFS1 interacts with SEC24A, SEC24B, and SEC24C. $n = 3$ independent experiments. **c** Representative imaging of the interaction between WFS1 and COPII vesicle subunits by PLA. The puncta number per cell was quantified by ImageJ software. $n = 3$ independent experiments, $n = 8$ independent images quantified. Scale bar, 10 μm. **d** In vitro pull-down analysis of the interaction between the N-terminal segment of WFS1 protein and COPII vesicle subunits. Purification of recombinant Sumo$^{His}$-tagged COPII vesicle subunits was subjected to pull down by the purified recombinant GST-tagged N-terminal segment of WFS1. IB analysis of pull-down samples showing the N-terminal segment of WFS1 interacts with SEC24A, SEC24B and SEC24C. $n = 3$ independent experiments. **e** Representative imaging of the interaction between SEC24A and mutant WFS1 proteins by PLA. The puncta number per cell was quantified by ImageJ software. $n = 3$ independent experiments, $n = 8$ independent images quantified. Scale bar, 10 μm. **f** In vitro pull-down analysis of the interaction between mutant WFS1 and the COPII vesicle subunit SEC24A. Lysates from HEK-293T cells expressing HA-tagged mutant WFS1 were subjected to pull down by the purified SEC24A protein. IB analysis of pull-down samples showing only WT WFS1 protein pulled down by SEC24A. $n = 3$ independent experiments. Molecular weights are in kDa. All the data are presented as mean ± s.e.m. $p < 0.05$, significant, using a two-tailed Student's $t$-test. Source data are provided as a Source Data file.

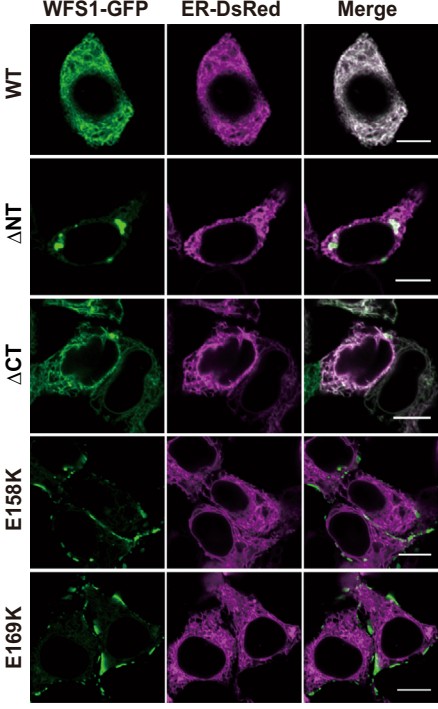

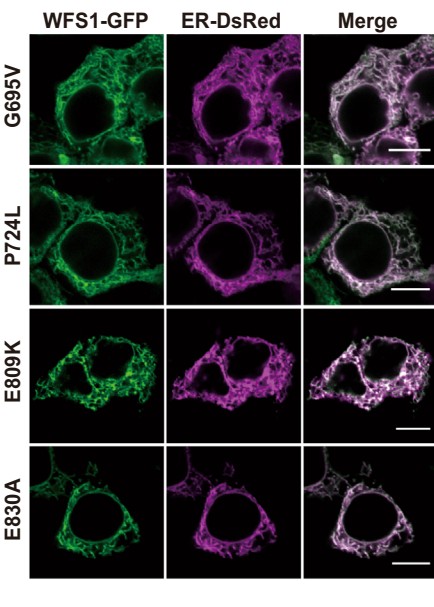

**Fig. 6 The localization of different mutant WFS1 proteins in HEK-293T cells.** Pairwise expression of KDEL-Dsred and mutant WFS1-GFP (WT, ΔNT, ΔCT, E158K, E169K, G695V, P724L, E809K, and E830A) revealed that the ΔNT, E158K, or E169K mutation severely disrupted ER localization of WFS1. $n = 3$ independent experiments. Scale bar, 10 μm.

granule cargo trafficking, which is a specific secretory pathway in pancreatic β-cells. Pathogenic mutations (E158K and E169K) in the N-terminal segment disrupt ER localization and its interaction with SEC24, and pathogenic mutations (G695V, P724L, E809K, and E830A) in the C-terminal segment disrupt the recognition of cargo proteins, impairing the sorting of cargo proteins into the COPII vesicles. Whereas blockade of COPII vesicle-mediated cargo trafficking induces ER stress and apoptosis in pancreatic β-cells[35]. Therefore, our data indicated that the impaired transport of vesicular cargo proteins underlies diabetes caused by *WFS1* pathogenic mutations. In addition to juvenile-onset diabetes, one of the major characteristics of Wolfram syndrome is neurodegeneration[9], and WFS1 has been shown to be highly expressed in the brain[14]. Thus, it will be of interest to verify whether WFS1 also acts as a receptor to sort neuropeptide cargo proteins into the COPII vesicles in neuronal cells. Recently, it has been reported that deficiency of YIPF5 in pancreatic β-cell models gave rise to proinsulin retention and ER stress[36], which is

quite similar to the phenotype caused by disrupting WFS1. YIPF5 resides in the ER and is assumed to be involved in ER-to-Golgi trafficking. It would be intriguing to test whether WFS1 interacts with YIPF5.

## Methods

**Cell culture and stable cell line**. INS1 cells were cultured in RPMI 1640 medium containing 10% fetal bovine serum (FBS), 1 mM sodium pyruvate, 50 μM β-mer-captoethanol, 100 U/mL penicillin, and 100 μg/mL streptomycin. Human embryonic kidney (HEK)-293T or platinum (Plat)-E cells were cultured in DMEM medium containing 10% FBS, 100 U/mL penicillin, and 100 μg/mL streptomycin. All the cells were placed in a humidified atmosphere with 5% $CO_2$ at 37 °C. Cell lines used in this study were not found in the BioSample database of commonly misidentified cell lines provided by the International Cell Line Authentication Committee (ICLAC).

The *Wfs1* knockdown INS1 cell line was constructed with sh*Wfs1* plasmids. The shRNA plasmid targeting *Wfs1* and a control shRNA plasmid (scrambled) were used to transfect INS1 cells with Lipofectamine 2000 (Invitrogen, Carlsbad, CA, USA), and positive cells were selected with G418 for 2 weeks. GFP-positive cells were sorted with flow cytometry to generate stable cell lines.

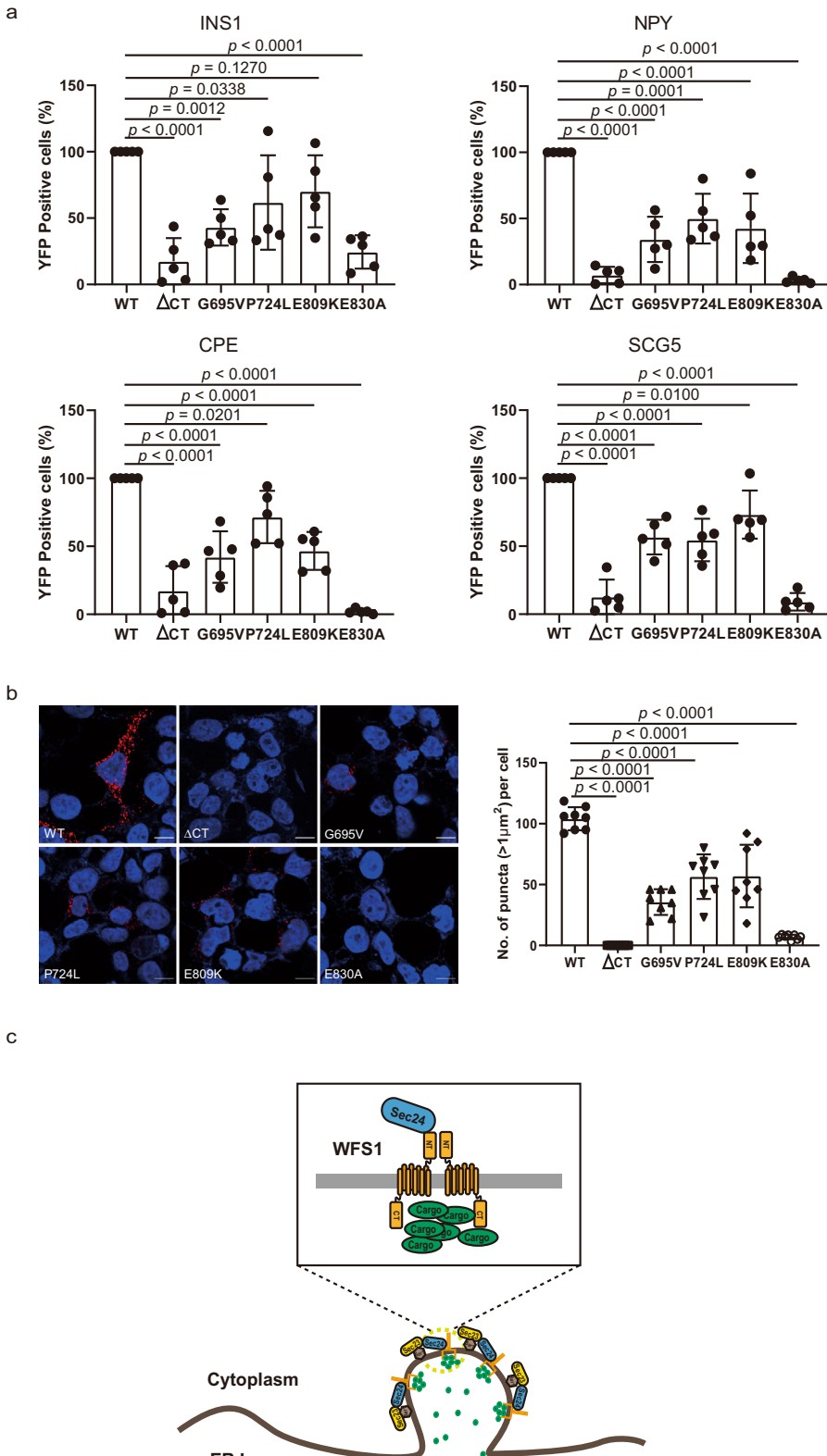

**Fig. 7 WFS1 recognizes vesicular cargo proteins via its luminal C-terminal segment. a** The interaction between different mutant WFS1 proteins and vesicular cargo proteins evaluated by BiFC and then quantified by flow cytometry. $n = 5$ independent experiments. **b** Representative imaging of the interaction between different mutant WFS1 and vesicular cargo protein SCG5 by PLA. The puncta number per cell was quantified by ImageJ software. $n = 4$ independent experiments, $n = 8$ independent images quantified. Scale bar, 10 μm. **c** A model of WFS1 mediating vesicular cargo protein transport from the ER. All the data are presented as mean ± s.e.m. $p < 0.05$, significant, using a two-tailed Student's $t$-test. Source data are provided as a Source Data file.

The WT and E830K mutant *Wfs1*-rescued cell lines were constructed with the retroviral pQCXIP plasmid in the sh*Wfs1* cells. Retrovirus was packaged by transfecting PlatE cells with HA-tagged *Wfs1* (WT and E830K mutation) plasmids and two packaging plasmids (vsvg and phit). At 48 h posttransfection, the virus-containing supernatant was collected, filtered and used to infect INS1 cells. The infected cells were selected and maintained with puromycin to generate stable cell lines.

**Animals.** *Wfs1* KO mice were generated by GemPharmatech using the CRISPR-Cas9 system. Two sgRNAs directed Cas9 endonuclease cleavage in intron 1 and exon 8 were designed by the online tool CRISPOR (http://crispor.tefor.net/), resulting in WFS1 frame mutations. Cas9 mRNA and sgRNAs were coinjected into the fertilized eggs of C57BL/6JGpt mice. The fertilized eggs were transplanted to form positive F0 mice, which were confirmed by PCR and Sanger sequencing. A 16,011 bp deletion at chromosome 5: 37,123,888–37,139,898 was detected, resulting in the *Wfs1* frame mutation. A stable F1 generation mouse model was obtained by mating positive F0 generation mice with WT C57BL/6JGpt mice. The top 3 off-target hits predicted by the online tool CRISPOR were checked by PCR and Sanger sequencing, and no off-target indels were found (see Supplemental Table 1 for primer sequence) in the homozygous *Wfs1* KO mice. The mice were housed in groups of three to five at 22–24 °C and 40–60% humidity, with a 12 h light/12 h dark cycle. Animals had access to water ad libitum. All mice used in this study were sex- and age-matched. All animal experiments were approved by the Animal Care Committee at the Institute of Biophysics (License No. SYXK2016-19).

**Molecular biology and constructs.** Total RNA was extracted from INS1 cells using TRIzol (Invitrogen, Carlsbad, CA, USA) as per the manufacturer's instructions. cDNA libraries were generated by retrotranscription of RNAs from INS1 cells using SuperRT reverse transcriptase (CWBio, Beijing, China) as per the manufacturer's instructions. Then, cDNA libraries were used to PCR-amplify cDNAs for vesicular cargo genes[37], *Wfs1*, *Tmed9*, *Sec24a*, *Sec24b*, *Sec24c*, *Sec23a* and *Sec23b*. With regard to the BiFC plasmids, cDNAs were inserted into the pcDNA3.1-nYFP and pcDNA3.1-cYFP constructs at HindIII and BamHI sites. pcDNA3.1-nYFP and pcDNA3.1-cYFP constructs were generated through the insertion of the N-terminus (1-154) and C-terminus (155-238) of YFP into the pcDNA3.1 vectors at the BamHI and EcoRI sites. For generation of the *Wfs1* mutants, WT *Wfs1* was used as a template for site-directed mutagenesis by over-lapping PCR. Concerning the plasmids expressed in *Escherichia coli* (*E. coli*), the cDNAs of *Sec24a*, *Sec24b*, *Sec24c*, *Sec23a* and *Sec23b* were subcloned into the BamHI and XhoI sites of a modified pET-28a vector (Novagen) with an N-terminal Sumo$^{His}$ tag, and the cDNA of the N-terminus of *Wfs1* (aa 32-285) was subcloned into the BamHI and XhoI sites of the pGEX-6p-1(GE Healthcare, Piscataway, NJ, USA) vector with an N-terminal GST-tag. All the oligonucleotides used for cloning are reported in Supplementary Table 2.

**IB.** Whole-cell lysates prepared using RIPA buffer containing 1% proteinase inhibitor (Sigma-Aldrich, St. Louis, MO, USA) were separated by sodium dodecyl sulfate-polyacrylamide gel electrophoresis (SDS-PAGE) and transferred onto PVDF membranes (Millipore, Billerica, MA, USA). The membranes were incubated with appropriate primary antibodies, followed by the appropriate HRP-conjugated secondary antibodies. Primary and HRP-conjugated secondary antibodies are reported in Supplementary Table 3. The protein expression was detected with enhanced luminescence reagents (GE Healthcare, Piscataway, NJ, USA).

**BiFC analysis.** HEK-293T cells were plated in 24-well plates for 24 h, then transfected with 200 ng nYFP- and 200 ng cYFP-tagged constructs. After 48 h, the cells treated with low temperature (30 °C) for 6 h for fluorophore maturation. Fluorescence was determined by flow cytometry using BD FACSCalibur (BD Biosciences) or imaged using a confocal laser scanning microscope (FV1200, Olympus, Tokyo, Japan).

**Immunofluorescence detection of insulin and proinsulin distribution.** WT, *Wfs1*-knockdown, or -rescued (WT or E830 mutation) *Wfs1* INS1 cells were grown on glass coverslips for 48 h, washed with phosphate-buffered saline (PBS) and fixed with 4% paraformaldehyde (PFA; Sigma-Aldrich, St. Louis, MO, USA) for 15 min. After washed with PBS for three times, the cells were permeabilized with 0.1% Triton X-100 in PBS for 10 min, then blocked with 5% goat serum for 1 h. Coverslips were then incubated with appropriate primary antibodies at room temperature for 1 h, and with Alexa-Fluor-conjugated secondary antibodies for 1 h. Primary and secondary antibodies are reported in Supplementary Table 3. Coverslips were mounted on glass slides with Vectashield DAPI (Invitrogen, Carlsbad, CA, USA) for nuclear staining. Fluorescence images were acquired using a confocal laser scanning microscope (FV1200, Olympus, Tokyo, Japan).

Pancreata from the WT and *Wfs1* KO mice were fixed with 4% PFA at 4 °C for 24 h. The fixed tissues were soaked sequentially in solutions of 100%, 95%, 85%, and 70% alcohol and then in dimethylbenzene; the tissues were then embedded in paraffin. First, 5 μm sections were prepared with a microtome. Then the sections were pretreated using a heat-mediated antigen retrieval buffer with sodium citrate (pH 6.0, epitope retrieval solution) for 20 min and were then blocked with 5% goat serum. The sections were incubated with appropriate primary antibodies at 4 °C overnight, followed by incubation with secondary antibodies and DAPI (Invitrogen, Carlsbad, CA, USA) for nuclear staining. Primary and secondary antibodies are reported in Supplementary Table 3. Fluorescence images were acquired using a confocal laser scanning microscope (FV1200, Olympus, Tokyo, Japan).

**Subcellular fractionation.** Two 10 cm dishes of INS1 cells were washed three times with ice-cold PBS and scraped in 1 mL of ice-cold homogenization buffer (250 mM sucrose, 1 mM EDTA, 0.03 mM cycloheximide, 3 mM imidazole, pH 7.4, and 1% proteinase inhibitor). The cells were homogenized with a 1 mL syringe attached to a 22 G needle, and then centrifuged at 2000 *g* for 10 min at 4 °C to remove nuclear debris and unbroken cells. The supernatant was gently retrieved and floated on a 10–45% continuous sucrose gradient in a SW41 tube. The samples were then centrifuged at 210,000 *g* for 18 h at 4 °C. The fractions were collected and concentrated by TCA precipitation for the subsequent IB analysis.

**Co-IP.** The cells were lysed in buffer A containing 20 mM Tris-HCl pH 7.5, 100 mM KCl, 5 mM MgCl₂, 0.5% NP-40, and 1% proteinase inhibitor for 30 min on ice and centrifuged at 10,000*g* for 15 min at 4 °C. The supernatants were incubated with protein A/G magnetic beads coated with an anti-HA antibody or control IgG (Beyotime, Shanghai, China) overnight at 4 °C. Then, the beads were washed three times with NT2 buffer (50 mM Tris-HCl pH 7.5, 150 mM NaCl, 1 mM MgCl₂, and 0.05% NP-40). The precipitates were incubated in SDS buffer containing 0.5% β-ME at 95 °C for 15 min and analyzed by IB.

**PLA.** The PLA assay was performed using Duolink In Situ PLA kit (Sigma-Aldrich, St. Louis, MO, USA) according to the manufacturer's instructions. Briefly, HEK-293T cells were pairwise cotransfected with HA-tagged *Wfs1* and vesicular cargo protein-tagged cYFP plasmids or HA-tagged *Wfs1* and COPII complex protein-tagged Myc plasmids for 48 h. The cells were fixed with 4% PFA for 15 min and permeabilized with 0.1% Triton X-100 in PBS for 10 min, then incubated with the appropriate primary antibodies against the two targets at 4 °C overnight. Primary antibodies are reported in Supplementary Table 3. After washing with PBS, Duolink proximity probes anti-rabbit PLUS and anti-mouse MINUS PLA probes targeting the primary antibodies were added to cells followed by ligation and amplification using the recommended conditions according to the manufacturer's instructions. Image acquisition was performed using a confocal laser scanning microscope (FV1200, Olympus, Tokyo, Japan). Quantification of the PLA signal was carried out using ImageJ.

**Protein expression and purification.** *E. coli* Rosetta BL21 (DE3) cells expressing Sumo$^{His}$-tagged Sec23/24 or the GST-tagged N-terminus of WFS1 were cultured with Terrific Broth (TB) medium at 37 °C and induced by 0.2 mM isopropyl β-D-1-thiogalactopyranoside (IPTG) at 16 °C for 16–18 h. When the OD600 reached 2.0, the cells were harvested and sonicated in PBS buffer with 400 mM NaCl and 0.01% Triton X-100, and then centrifuged at 32,000*g* for 30 min at 4 °C. The supernatants were incubated with Ni-NTA beads (GE Healthcare, Piscataway, NJ, USA) or 4B GST beads (GE Healthcare, Piscataway, NJ, USA) for Sumo$^{His}$-tagged proteins or GST-tagged proteins for 1 h, respectively. Then, washed with PBS for six times and eluted by PBS with 300 mM Imidazole. The eluted proteins were dialyzed against buffer B (20 mM Tris-HCl pH 8.0, 300 mM NaCl, and 0.5 mM EDTA) and stored at −80 °C for the subsequent binding assay.

**In vitro binding assays.** In all, 10 μg purified fusion proteins (Sumo$^{His}$-SEC24A, Sumo$^{His}$-SEC24B, Sumo$^{His}$-SEC24C, Sumo$^{His}$-SEC23A, and Sumo$^{His}$-SEC23B) in buffer B with 10 mM imidazole were bound to pre-equilibrated and pre-chilled Ni beads for 30 min at 4 °C, and then washed with buffer C (20 mM Tris-HCl pH 8.0, 500 mM NaCl, 1 mM β-ME, and 0.02% Triton X-100) three times to remove the unbound proteins. HEK-293T cells expressing HA-tagged WFS1 were lysed with buffer A for 30 min on ice and centrifuged at 10,000*g* for 15 min at 4 °C. The supernatants were incubated with protein-bound Ni beads at 4 °C for 1 h, then washed three times with buffer C. The precipitates were incubated in SDS buffer containing 0.5% β-ME at 95 °C for 30 min and analyzed by IB. Sumo-tag-bound beads were used as controls.

In all, 10 μg purified GST-fused WFS1 N-terminal segment in buffer D (20 mM Tris-HCl pH 8.0, 300 mM NaCl, 0.5 mM EDTA, and 1 mM DTT) was bound to pre-equilibrated and pre-chilled GST beads for 30 min at 4 °C, washed with buffer C three times to remove the unbound proteins. The protein-bound GST beads were incubated with purified Sumo$^{His}$-tagged proteins (Sumo$^{His}$-SEC24A, Sumo$^{His}$-SEC24B, Sumo$^{His}$-SEC24C, Sumo$^{His}$-SEC23A, and Sumo$^{His}$-SEC23B) at 4 °C for 1 h, and then washed with buffer C three times. The precipitates were incubated in SDS buffer containing 0.5% β-ME at 95 °C for 30 min and analyzed by IB. GST-tag-bound beads were used as controls.

**Islet isolation.** The pancreas was inflated by instilling 5 mL of Hanks' buffered saline solution (HBSS) containing 0.5 mg/mL collagenase P (Roche Applied Science, Mannheim, Germany) through the pancreatic duct. The pancreas was

harvested and incubated in a water bath at 37 °C for 25 min. The digested pancreas was rinsed with HBSS, and islets were separated on a Ficoll density gradient. After three washes with HBSS, the islets were manually isolated under a dissection microscope.

**In vitro insulin release assay**. Isolated islets were cultured in RPMI 1640 medium with 5.6 mM glucose for 24 h. After washed with PBS, 20 islets from each WT and KO group, were pre-incubated for 1 h in 2.8 mM glucose Krebs-Ringer bicarbonate HEPES buffer (KRBB) containing: 114 mM NaCl, 4.7 mM KCl, 1.2 mM $KH_2PO_4$, 1.16 mM $MgSO_4$, 0.5 mM $MgCl_2$, 2.5 mM $CaCl_2$, and 20 mM HEPES with 0.2% BSA, pH 7.4. Next, groups of islets were batch-incubated in 0.2 mL of 2.8 mM glucose in KRBB for 1 h. Incubation medium was withdrawn for insulin measurement after gentle agitation and was replaced by 0.2 mL of fresh KRBB solution supplemented with 16.8 mM glucose. All operations were conducted under dissection microscopy to avoid damaging the islets. Insulin was quantified by ELISAs with commercially available kits (Shibayagi, Shibukawa, Japan).

**Statistical analysis**. All statistical analyses were acquired from more than three independent experiments, and values were expressed as mean ± s.e.m. Significance was calculated by two-tailed paired Student's $t$-tests. A $p$ value of less than 0.05 was considered statistically significant. Statistical analyses were performed in GraphPad Prism.

**Reporting summary**. Further information on research design is available in the Nature Research Reporting Summary linked to this article.

## Data availability
All data supporting the findings of this study. Source data are provided with this paper.

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

## Acknowledgements
We thank Dr. Pingyong Xu for critical reading of the manuscript and for advice. We thank the staff of the Institute of Biophysics Core Facilities, in particular, Junying Jia for technical support with flow cytometry, Yan Teng for technical support with confocal imaging, Zheng Liu for technical support with mouse feeding. This work was supported by grants from the National Key Research and Development Program (Grant No. 2016YFA0500203 to T.X.), the National Natural Science Foundation of China (Grant No. 31730054 to T.X.), and the Strategic Priority Research Program of the Chinese Academy of Sciences (Grant No. XDA12030101, XDA12040104 to Y.W.).

## Author contributions
Z.L. and T.X. initiated and designed the project. Z.L., L.W., and T.X. prepared the manuscript. Z.L. and L.W. made the constructs and performed the BiFC experiments. L.W. carried out most of the experimental work, including the mouse experiments, PLA, microscope analysis, protein purification, pull down, and IB. H.L. and X.Z. performed the Co-IP experiments. Y.W. and E.S. reviewed and edited the manuscript. Y.W. and T.X. received the grants.

## Competing interests
The authors declare no competing interests.
