## [Peer Review File. · Nature Communications]

WFS1 functions in ER export of vesicular cargo proteins in pancreatic β -cells

Editorial Note: Parts of this Peer Review File have been redacted as indicated to maintain the confidentiality of unpublished data.REVIEWER COMMENTS

Reviewer #1 (Remarks to the Author):

This is an interesting and very clearly written report which is adding to the existing efforts to try and understand the multiple functions of the WFS1 gene, which is known to cause a spectrum of disease including neonatal and early onset diabetes. I have listed my comments and suggestions below:

1. CRISPR – the methods sections should include details on how the gRNAs were designed and whether any off target effect was predicted in silico. It would also be useful to provide details of the deletion generated (such as chromosome coordinates). Was the deletion homozygous in the mice that were further assessed? Also, did the authors conduct any assays to exclude off target effects?

2. Mouse KO – could the authors comment on how similar the Wfs1 KO mouse was phenotypically to those previously described? Beta cell morphology abnormalities have been previously reported in Wfs1 KO mouse, was this observed in this experiment as well? Although I realise the space is limited, even just a sentence highlighting consistency (or not) with the previous models would be useful the readers.

3. The figure references in the first section of results are confusing, the authors only refer to Figure 1 but from the legend I gather that Figure 1 represents the in vitro results and Figure 2 (which is not mentioned anywhere in the manuscript) the KO mouse data? At the end of page 5 the authors mention figure 1i-l – I can't find figure 1l, should this be figure 2i-l?

4. Could the authors comment on why they have used the human T293 cells for the proinsulin interaction analyses and WFS1 trafficking experiments rather than using the rat INS1 cells as they did for the proinsulin experiment? Was it because of reduced transfection efficiency in INS1 cells? If that is the case, can the authors comment on why they did not choose another rodent line?

5. Can the authors comment on whether they see any differences in localisation patterns of the proteins with the four pathogenic mutations (G695V, P724L, E809K and E830A) compared to the results in COS7 cells in Ref 11? Protein misfolding resulting in ER stress has been suggested as a possible mechanism, especially for the E809K and E830A variants which differ from the others as they cause a congenital subtype of WFS1 disease which is dominantly caused by de novo mutations. Can the authors comment on whether they see any evidence of misfolding and whether there was any difference between the recessive and dominant mutations?

6. A recent report showed proinsulin retention in the ER in beta cells resulting from YIPF5 loss of function/partial loss of function. Mutations in YIPF5 (which is presumed to be involved in ER to Golgi trafficking) also cause diabetes? Do the authors think these two disease genes may be resulting in diabetes through a similar mechanism?

Minor point:

- The use of the gene nomenclature is inconsistent, the gene name should be in lower case with capital first letter and italics when referring to the mouse gene and all upper case and italics for the human gene

- Line 25: different font

Reviewer #2 (Remarks to the Author):

In this manuscript, Wang et al. make the important discovery that the diabetes-associated WFS1 protein acts as an ER-to-Golgi cargo receptor for proinsulin, the protein precursor for insulin.

The authors start with observations in INS1 cells (an established model for insulin secretion) showing that knockdown of WFS1 results in defective maturation of proinsulin to insulin, with marked retention of proinsulin in the ER and resulting diminished proinsulin in the Golgi. Similar results are obtained in the pancreas of a Wfs1 KO mouse line generated by the authors and displaying various phenotypes associated with diabetes.

Assays based on bimolecular fluorescence complementation, proximity ligation, and immunoprecipitation show that WFS1 interacts with proinsulin (and about half of other soluble cargoes tested). Subsequent experiments show that WFS1 traffics from the ER to the Golgi and plasma membrane and, notably, identify two motifs at the N-terminus of WFS1 that are involved in the interaction of WFS1 with specific COPII subunits for WFS1 ER export. Finally, the authors demonstrate that the luminal C-terminus of WFS1 mediates interaction with its cargo proteins and that this interaction is diminished or disrupted by known pathogenic variants.

The story is compelling and the experiments straightforward and well executed/controlled. The significance of the manuscript is high in that only a handful of mammalian ER cargo receptors have been characterized so far, and notably, this one has relevance for a pathway as important as that of insulin metabolism.

I just have a few very minor observations which, if addressed, could help clarify some mechanistic details:

1. What the authors think could be an explanation for the observation that disruption of either WFS1 ER export signal results in WFS1 localization at the plasma membrane? Is mutant WFS1 following a route that is independent of COPII vesicles?

2. In the BiFC assay, complex reconstitution is known to be irreversible. Given the apparent robust signal displayed in the interaction of WFS1 with proinsulin (and other cargoes as well), could this property be used to track where the reconstituted complex goes when the ER export signals are mutated? Given that pathogenic mutations have been mapped to these signals--but still, WFS1 appears to be able to leave the ER---this could provide hints on the fate of the cargo proteins in these patients. An alternative or complementary experiment could be performed by expressing the mutant WFS1 constructs in the available Wfs1 knock-out cells to again monitor the distribution of proinsulin.

3. Page 5, "indicating the proinsulin can not be delivered to Golgi complex for processing" should be modified – the data show that the delivery of proinsulin to the Golgi is severely impaired, not completely eliminated.

4. The callout of Fig. 1c should be placed before of the callout for Fig. 1d and following.

5. The callouts of Fig. 2a-h are erroneously indicated as Fig.1a-h in the text.

6. Pag. 6, line 6. Given that the BiFC assay employs exogenously expressed proteins, the word "endogenous" should be modified or eliminated.

Response to the reviewers

We appreciate you and the reviewers for your precious time in reviewing our paper and providing valuable comments. It was your valuable and insightful comments that led to possible improvements in the current version. We have carefully considered the comments and tried our best to address every one of them. We hope the manuscript after careful revisions meet your high standards. The authors welcome further constructive comments if any.

Below we provide the point-by-point responses. All modifications in the new manuscript have been highlighted in red.

Reviewer #1 (Remarks to the Author):

This is an interesting and very clearly written report which is adding to the existing efforts to try and understand the multiple functions of the WFS1 gene, which is known to cause a spectrum of disease including neonatal and early onset diabetes. I have listed my comments and suggestions below:

1. CRISPR – the methods sections should include details on how the gRNAs were designed and whether any off target effect was predicted in silico. It would also be useful to provide details of the deletion generated (such as chromosome coordinates). Was the deletion homozygous in the mice that were further assessed? Also, did the authors conduct any assays to exclude off target effects?

Response: We thank the reviewer for the constructive suggestions. We have described how to design the sgRNAs in the methods sections in detail.

Two sgRNAs directed cas9 endonuclease cleavage in intron 1 and exon 8 of *Wfs1* were designed by the online tool CRISPOR (<http://crispor.tefor.net/>). The on-target locus and top 3 off-target hits predicted by the online tool CRISPOR were checked by the PCR and Sanger sequencing. The results showed a 1601bp deletion at Chromosome 5: 37,123,888-37,139,898, resulting in WFS1 frame mutate and no

off-target indels were found (see Supplemental Table 3 for primer sequence) in homozygous *Wfs1* KO mice.

2. Mouse KO – could the authors comment on how similar the *Wfs1* KO mouse was phenotypically to those previously described? Beta cell morphology abnormalities have been previously reported in *Wfs1* KO mouse, was this observed in this experiment as well? Although I realise the space is limited, even just a sentence highlighting consistency (or not) with the previous models would be useful the readers.

Response: We thank the reviewer's suggestion. We have compared the phenotype between our *Wfs1* KO mouse and previously reported *Wfs1* KO mouse. As the reviewer pointed out, the β -cells and islet morphology were abnormal, which are consistent with previous models. As suggested by the reviewer, we have added the following sentence in the results of revised manuscript.

Page 5, line 92, “Besides, *Wfs1* KO mice exhibited an impaired glucose tolerance as well as diminished islet size and abnormal islet morphology (Supplementary Fig.1d-g), which are consistent with previous models^{16,17}”

3. The figure references in the first section of results are confusing, the authors only refer to Figure 1 but from the legend I gather that Figure 1 represents the in vitro results and Figure 2 (which is not mentioned anywhere in the manuscript) the KO mouse data? At the end of page 5 the authors mention figure 1i-1 – I can't find figure 1i, should this be figure 2i-1?

Response: We are really sorry for these mistakes. The figure 1i-1 should be figure 2i-1. We have corrected the mis-refered figures in the results section of revised manuscript.

4. Could the authors comment on why they have used the human T293 cells for the proinsulin interaction analyses and WFS1 trafficking experiments rather than using the rat INS1 cells as they did for the proinsulin experiment? Was it because of reduced transfection efficiency in INS1 cells? If that is the case, can the authors comment on why they did not choose another rodent line?

Response: We thank the reviewer for the suggestions. We re-performed the interaction and WFS1 trafficking experiments in INS1 cells. As the reviewer pointed

out, the transfection efficiency is low in INS1 cells. Even in this circumstance, we can still observe the BiFC positive signals when pair-transfection of WFS1-nYFP and cargo protein-cYFP (Supplementary Fig. 3a). But the efficiency of positive signal is too low to analyze statistically. Moreover, the interaction of WFS1 with endogenous cargo proteins were further confirmed by immunoprecipitation in INS1 cells (Supplementary Fig. 3b). For the WFS1 trafficking experiments, a low temperature treatment increases the localization of WFS1 to the Golgi complex in INS1 cells (Supplementary Fig. 4), consistence with the results in 293T cells. In addition, the previous studies showed that the WFS1 localizes to secretory granules in neuronal cells and pancreatic β -cells (Gharanei et al., 2013; Hatanaka et al., 2011), further confirming that WFS1 is a trafficking protein.

Supplementary Fig. 3: WFS1 interacts with vesicular cargo proteins in INS1 cells. **a**, Representative live imaging of reconstituted BiFC fluorescence between WFS1-tagged nYFP or nYFP-tagged TMED9, and vesicular cargo proteins (INS1, NPY, CPE, SCG5 and negative control GCG)-tagged cYFP. Green fluorescence shows reconstitution of YFP as an indicator of protein– protein interaction. Images are representative of n = 3 independent

experiments. Scale bar, 10 μm . **b**, IP analysis of the interaction of WFS1 with endogenous vesicular cargo proteins in INS1 cells. The immunoprecipitates pulled down by the WFS1 antibody were analyzed by IB with the indicated antibodies. GCG was used as a negative control. Input represents 5% of the total cell extract used for immunoprecipitation. Molecular weights are in kDa. Images are representative of $n = 3$ independent experiments.

Supplementary Fig. 4: WFS1 traffics from ER to Golgi in INS1 cells. a-d, Confocal microscope analysis of co-localization of WFS1-GFP and ER-Dsred in INS1 cells treated with normal (37 °C) or low temperature (30 °C) (**a**). Trace outline is used for line-scan (white dashed line) analysis of relative fluorescence intensity of WFS1 and ER signals treated with normal (37 °C, **b**) or low temperature (30 °C, **c**). Signal overlap is quantified by Pearson correlation analysis of $n = 3$ independent experiments (**d**). Data are means \pm s.e.m. (***) $P < 0.001$, two-tailed Student's t-test). Scale bar, 10 μm . **e-h**, Confocal microscope analysis of co-localization of WFS1-GFP and

Golgi-TDimer2 in HEK-293T cells treated with normal (37 °C) or low temperature (30 °C) (**e**). Trace outline is used for line-scan (white dashed line) analysis of relative fluorescence intensity of WFS1 and Golgi signals treated with normal (37 °C, **f**) or low temperature (30 °C, **g**). Signal overlap is quantified by Pearson correlation analysis of n = 3 independent experiments (**h**). Data are means \pm s.e.m. (**P < 0.01, ***P < 0.001, two-tailed Student's t-test). Scale bar, 10 μ m.

5. Can the authors comment on whether they see any differences in localisation patterns of the proteins with the four pathogenic mutations (G695V, P724L, E809K and E830A) compared to the results in COS7 cells in Ref 11? Protein misfolding resulting in ER stress has been suggested as a possible mechanism, especially for the E809K and E830A variants which differ from the others as they cause a congenital subtype of WFS1 disease which is dominantly caused by de novo mutations. Can the authors comment on whether they see any evidence of misfolding and whether there was any difference between the recessive and dominant mutations?

Response: We thank the reviewers for the comments. The difference between the recessive and dominant mutations is also the focus of our study. We compared the localization of WFS1 proteins with the four pathogenic mutations (G695V, P724L, E809K and E830A) in 293T cells and COS7 cells in Ref 11 (De Franco et al., 2017). This study reported that both dominant and recessive WFS1 mutants (G695V, P724L, E809K and E830A) showed a puncta pattern in the ER of COS7 cells, suggesting a tendency for these WFS1 mutants to misfold and aggregate. However, only dominant mutations showed puncta pattern in the ER of INS1 cells. We also performed the similar experiments in 293T cells. No obvious puncta pattern was observed in 293T cells.

[REDACTED]

6. A recent report showed proinsulin retention in the ER in beta cells resulting from YIPF5 loss of function/partial loss of function. Mutations in YIPF5 (which is presumed to be involved in ER to Golgi trafficking) also cause diabetes? Do the authors think these two disease genes may be resulting in diabetes through a similar mechanism?

Response: We thank the reviewers for this interesting comments. The physiological function, expression pattern in tissues, and localization of WFS1 and YIPF5 are similar (De Franco et al., 2020). For example, mutation of either WFS1 or YIPF5 can cause neonatal diabetes. Both of them are ER-associated membrane proteins, and are highly expressed in pancreatic β cells and brain. Loss of function/partial loss of function of either WFS1 or YIPF5 in pancreatic β cells resulted in proinsulin retention in the ER, increased ER stress response and β cells failure. Therefore, we think these two disease genes may be resulting in diabetes through a similar mechanism.

Except for these two disease genes, there is another example that two disease genes, CLN6 and CLN8, result in Batten disease classified as a lysosomal storage disorder (Bajaj et al., 2020; di Ronza et al., 2018). Mechanistically, CLN8 as a receptor interacts with newly synthesized lysosomal enzymes, transfers them to the Golgi via COPII vesicles. Whereas CLN6 is not loaded into COPII vesicles but is retained in the ER, presumably to serve additional cycles of enzyme recruitment. CLN6 and CLN8 are obligate partner for the recruitment of newly synthesized lysosomal enzymes from ER to COPII vesicles. Therefore, it is intriguing to verify whether the WFS1 and YIPF5 can form an obligate partner? And whether there is interaction between the YIPF5 and vesicular cargo proteins?

We have added a sentence in the discussion as follows: “Recently, it has been reported that deficiency of YIPF5 in pancreatic β -cells models gave rise to proinsulin retention and ER stress (De Franco et al., 2020), which is quite similar to the phenotype caused by disrupting WFS1. YIPF5 resides in the ER and is assumed to be involved in ER to Golgi trafficking. It would be intriguing to test whether WFS1 interacts with YIPF5.

Minor point:

- The use of the gene nomenclature is inconsistent, the gene name should be in lower case with capital first letter and italics when referring to the mouse gene and all upper case and italics for the human gene

Response: Thank you for the nice reminder. We went through the entire manuscript to eliminate gene nomenclature inconsistent.

- Line 25: different font

Response: We have made revisions accordingly.

Reviewer #2 (Remarks to the Author):

In this manuscript, Wang et al. make the important discovery that the diabetes-associated WFS1 protein acts as an ER-to-Golgi cargo receptor for proinsulin, the protein precursor for insulin.

The authors start with observations in INS1 cells (an established model for insulin secretion) showing that knockdown of WFS1 results in defective maturation of proinsulin to insulin, with marked retention of proinsulin in the ER and resulting diminished proinsulin in the Golgi. Similar results are obtained in the pancreas of a Wfs1 KO mouse line generated by the authors and displaying various phenotypes associated with diabetes.

Assays based on bimolecular fluorescence complementation, proximity ligation, and immunoprecipitation show that WFS1 interacts with proinsulin (and about half of other soluble cargoes tested). Subsequent experiments show that WFS1 traffics from the ER to the Golgi and plasma membrane and, notably, identify two motifs at the N-terminus of WFS1 that are involved in the interaction of WFS1 with specific COPII subunits for WFS1 ER export. Finally, the authors demonstrate that the luminal C-terminus of WFS1 mediates interaction with its cargo proteins and that this interaction is diminished or disrupted by known pathogenic variants.

The story is compelling and the experiments straightforward and well executed/controlled. The significance of the manuscript is high in that only a handful of mammalian ER cargo receptors have been characterized so far, and notably, this

one has relevance for a pathway as important as that of insulin metabolism.

I just have a few very minor observations which, if addressed, could help clarify some mechanistic details:

1. What the authors think could be an explanation for the observation that disruption of either WFS1 ER export signal results in WFS1 localization at the plasma membrane? Is mutant WFS1 following a route that is independent of COPII vesicles?

Response: We thank the reviewer for the comments. The reviewer raises a very interesting question. We agree with the reviewer's opinion that disruption of either WFS1 ER export signal results in WFS1 mislocalization at the plasma membrane via a COPII-independent pathway. There are a few of examples that transport to plasma membrane via a COPII-independent pathway (namely Golgi bypass), including CD45, hemichannel proteins, CFTR and *Drosophila* α PS1 integrin (Grieve and Rabouille, 2011). The Golgi bypass pathway is characterized with EndoH sensitivity, brefeldin A resistance and Golgi SNARE independence. CD45 reaches the plasma membrane in two differentially glycosylated forms, one EndoH-resistant (classical) and one EndoH-sensitive that potentially bypasses the Golgi. Therefore, it is intriguing to test whether the ER export signal mutant WFS1 is EndoH sensitivity or brefeldin A resistance? Our preliminary results showed that the trafficking of WFS1 is brefeldin A resistance. We will address this question in the following studies. In addition, the detail molecular mechanism regarding the Golgi bypass is largely unknown. Recently, Liang Ge group identified the TMED10, which is required for the unconventional protein secretion (Zhang et al., 2020). We will compare the interaction proteins between the WT-WFS1 and ER export signal mutant WFS1. We hope to identify the proteins that are required for the ER export signal mutant WFS1 trafficking, and illustrate the molecular mechanism regarding the Golgi bypass.

2. In the BiFC assay, complex reconstitution is known to be irreversible. Given the apparent robust signal displayed in the interaction of WFS1 with proinsulin (and other cargoes as well), could this property be used to track where the reconstituted complex goes when the ER export signals are mutated? Given that pathogenic mutations have been mapped to these signals--but still, WFS1 appears to be able to leave the ER—this could provide hints on the fate of the cargo proteins in these patients. An alternative or complementary experiment could be performed by expressing the mutant WFS1 constructs in the available Wfs1 knock-out cells to again monitor the distribution of proinsulin.

Response: We thank the reviewer for the constructive suggestion. As the reviewer

pointed out, the robust and irreversible signal of the BiFC assay is suitable for track where the cargoes go. We performed the experiment and the results showed the reconstituted complex of WT-WFS1 and cargo protein mainly localized in the Golgi complex after low temperature treatment as anticipated (below figure). However, very weak and less BiFC signal could be detected when co-transfection of the ER export signal (E158K or E169K) mutant WFS1 with cargo proteins, because the ER export signal mutations disrupt the ER localization of WFS1 and induce the physical distance between the WFS1 and cargo proteins. Therefore, this method could not be used to monitor the cargo distribution in these patients. As reviewer's suggestion, we perform the alternative experiment by expressing the E830K mutant WFS1 constructs in the *Wfs1* knock-down cells (sh*Wfs1*) to again monitor the distribution of proinsulin. The results showed that the proinsulin mainly localized to the Golgi in WT-WFS1 rescued cells. However, the proinsulin in E830K mutant WFS1 rescued cells mainly localized to the ER, not with the Golgi, which is similar to sh*Wfs1* INS1 cells (Supplementary Fig. 6). These results suggested that the mutant WFS1 could not rescue the proinsulin distribution in sh*Wfs1* INS1 cells.

WFS1-nYFP with SCG5-cYFP

Golgi

Merge

Figure legend: Low temperature treatment induce the reconstituted complex of WFS1 and SCG5 localized in the Golgi complex.

Supplementary Fig. 6: The effect of E830A mutant WFS1 on proinsulin distribution. **a**, Western blot analysis of WFS1 protein in the scrambled (NC), *shWfs1*, rescued WT-*Wfs1* and E830A mutant *Wfs1* INS1 cells; β -actin was used as the loading control. **b-m**, Confocal microscope analysis of co-localization of proinsulin with calnexin (**b-g**) or GM130 (**h-m**). Trace outline is used for line-scan (white dashed line) analysis of relative fluorescence intensities of proinsulin with calnexin or GM130 signals. Signal overlap is quantified by Pearson correlation analysis. Data are means \pm s.e.m. (n = 3 independent experiments, n = 6 independent images quantified, ****P < 0.0001, two-tailed Student's t-test). Scale bar, 5 μ m.

3. Page 5, “indicating the proinsulin can not be delivered to Golgi complex for processing” should be modified – the data show that the delivery of proinsulin to the Golgi is severely impaired, not completely eliminated.

Response: We have made revisions accordingly.

Page 5, line 77, “Moreover, the knockdown of WFS1 caused a significant increased ratio of proinsulin to insulin compared with scrambled cells (Fig.1c), indicating that the delivery of proinsulin to the Golgi is severely impaired.”

4. The callout of Fig. 1c should be placed before of the callout for Fig. 1d and following.

Response: Thank you for the nice reminder. We have made revisions accordingly.

5. The callouts of Fig. 2a-h are erroneously indicated as Fig.1a-h in the text.

Response: Thank you for the nice reminder. We have made revisions accordingly.

6. Pag. 6, line 6. Given that the BiFC assay employs exogenously expressed proteins, the word “endogenous” should be modified or eliminated.

Response: We have made revisions accordingly.

Page 6, line 108, “We employed a bimolecular fluorescence complementation (BiFC) system based on a split yellow fluorescent protein (YFP) variant to test the protein interaction in live cells.”

References

Bajaj, L., Sharma, J., di Ronza, A., Zhang, P., Eblimit, A., Pal, R., Roman, D., Collette, J.R., Booth, C., Chang, K.T., et al. (2020). A CLN6-CLN8 complex recruits lysosomal enzymes at the ER for Golgi transfer. *J Clin Invest* 130, 4118-4132.

De Franco, E., Flanagan, S.E., Yagi, T., Abreu, D., Mahadevan, J., Johnson, M.B., Jones, G., Acosta, F., Mulaudzi, M., Lek, N., et al. (2017). Dominant ER Stress-Inducing WFS1 Mutations Underlie a Genetic Syndrome of Neonatal/Infancy-Onset Diabetes, Congenital Sensorineural Deafness, and Congenital Cataracts. *Diabetes* 66,2044-2053.

De Franco, E., Lytrivi, M., Ibrahim, H., Montaser, H., Wakeling, M.N., Fantuzzi, F., Patel, K., Demarez, C., Cai, Y., Igoillo-Esteve, M., et al. (2020). YIPF5 mutations cause neonatal diabetes and microcephaly through endoplasmic reticulum stress. *J Clin Invest* 130, 6338-6353.

di Ronza, A., Bajaj, L., Sharma, J., Sanagasetti, D., Lotfi, P., Adamski, C.J., Collette, J., Palmieri, M., Amawi, A., Popp, L., et al. (2018). CLN8 is an endoplasmic reticulum cargo receptor that regulates lysosome biogenesis. *Nat Cell Biol* 20, 1370-1377.

Gharanei, S., Zatyka, M., Astuti, D., Fenton, J., Sik, A., Nagy, Z., and Barrett, T.G. (2013). Vacuolar-type H⁺-ATPase V1A subunit is a molecular partner of Wolfram syndrome 1 (WFS1) protein, which regulates its expression and stability. *Hum Mol Genet* 22, 203-217.

Grieve, A.G., and Rabouille, C. (2011). Golgi bypass: skirting around the heart of classical secretion. *Cold Spring Harb Perspect Biol* 3.

Hatanaka, M., Tanabe, K., Yanai, A., Ohta, Y., Kondo, M., Akiyama, M., Shinoda, K., Oka, Y., and Tanizawa, Y. (2011). Wolfram syndrome 1 gene (WFS1) product localizes to secretory granules and determines granule acidification in pancreatic beta-cells. *Hum Mol Genet* 20, 1274-1284.

Tunyasuvunakool, K., Adler, J., Wu, Z., Green, T., Zielinski, M., Zidek, A., Bridgland, A., Cowie, A., Meyer, C., Laydon, A., et al. (2021). Highly accurate protein structure prediction for the human proteome. *Nature* 596, 590-596.

Zhang, M., Liu, L., Lin, X., Wang, Y., Li, Y., Guo, Q., Li, S., Sun, Y., Tao, X., Zhang, D., et al. (2020). A Translocation Pathway for Vesicle-Mediated Unconventional Protein Secretion. *Cell* 181, 637-652 e615.

REVIEWERS' COMMENTS

Reviewer #1 (Remarks to the Author):

The authors have addressed all the points raised and I have no further comments. I would like to congratulate the authors on this very interesting piece of work.

Reviewer #2 (Remarks to the Author):

The authors have satisfactorily addressed my observations.

Response to the reviewers

Reviewer #1 (Remarks to the Author):

The authors have addressed all the points raised and I have no further comments. I would like to congratulate the authors on this very interesting piece of work.

R: Thanks.

Reviewer #2 (Remarks to the Author):

The authors have satisfactorily addressed my observations.

R: Thanks.